# Omega-3 Fatty Acid Intake by Age, Gender, and Pregnancy Status in the United States: National Health and Nutrition Examination Survey 2003–2014

**DOI:** 10.3390/nu11010177

**Published:** 2019-01-15

**Authors:** Maranda Thompson, Nicholas Hein, Corrine Hanson, Lynette M. Smith, Ann Anderson-Berry, Chesney K. Richter, Karl Stessy Bisselou, Adams Kusi Appiah, Penny Kris-Etherton, Ann C. Skulas-Ray, Tara M. Nordgren

**Affiliations:** 1Department of Pediatrics, University of Nebraska Medical Center, Omaha, NE 68198, USA; maranda.thompson@unmc.edu (M.T.); alanders@unmc.edu (A.A.-B.); 2College of Public Health, University of Nebraska Medical Center, Omaha, NE 68198, USA; nicholas.hein@unmc.edu (N.H.); lmsmith@unmc.edu (L.M.S.); karlstessy.bisselou@unmc.edu (K.S.B.); adams.kusiappiah@unmc.edu (A.K.A.); 3College of Allied Health Professions, University of Nebraska Medical Center, Omaha, NE 68198, USA; ckhanson@unmc.edu; 4Department of Nutritional Sciences, University of Arizona, Tucson, AZ 85721, USA; richterck@email.arizona.edu (C.K.R.); skulasray@email.arizona.edu (A.C.S.-R.); 5Department of Nutritional Science, The Pennsylvania State University, University Park, PA 16802, USA; pmk3@psu.edu; 6Division of Biomedical Sciences, School of Medicine, University of California Riverside, Riverside, CA 92521, USA

**Keywords:** eicosapentaenoic acid, docosahexaenoic acid, omega-3 fatty acids, lifespan, oily fish, fish oil supplements

## Abstract

Despite the importance of n-3 fatty acids for health, intakes remain below recommended levels. The objective of this study was to provide an updated assessment of fish and n-3 fatty acid intake (i.e., eicosapentaenoic acid (EPA), docosahexaenoic acid (DHA), and EPA+DHA) in the United States using the 2003–2014 National Health and Nutrition Examination Survey (NHANES) data (*n* = 45,347)). Over this survey period, toddlers, children, and adolescents (aged 1–19) had significantly lower n-3 fatty acid intake (*p* < 0.001) compared to adults and seniors, which remained significant after adjusting for caloric intake. Females demonstrated lower n-3 fatty acid intake than males (*p* < 0.001), with adult and senior women having significantly lower intakes compared to men in the same age categories (*p* < 0.001) after adjustment for energy intake. Women also consumed less fish than men (5.8 versus 6.1 servings/month, *p* < 0.001). The estimated intakes of n-3 fatty acids in pregnant women did not differ from non-pregnant women (*p* = 0.6 for EPA+DHA), although pregnant women reported consuming less high n-3 fatty acid-containing fish than non-pregnant women (1.8 versus 2.6 servings/month, *p* < 0.001). Our findings indicate that subgroups of the population may be at higher risk of n-3 fatty acid intakes below recommended levels.

## 1. Introduction

The major n-3 polyunsaturated fatty acids, eicosapentaenoic acid (EPA) and docosahexaenoic acid (DHA), play key physiological roles related to health and disease, and the health benefits of n-3 fatty acids are well established [1,2,3,4,5,6,7,8,9]. Intake of fatty fish is the primary contributor of EPA and DHA in the diet, with α-linolenic acid (ALA) conversion serving as an additional, but limited, source [10]. Supplementation with fish, krill, and cod liver oils, as well as other products derived from algal sources can also improve n-3 fatty acid status [11]. 

The health benefits conferred by n-3 fatty acids are important throughout the lifespan. In early life, n-3 fatty acids play an essential role in growth and development, including development of the eyes and brain [12]. Increased intake of n-3 fatty acids during pregnancy has been associated with decreased maternal depression [13], reduced rates of intrauterine growth restriction [12], preterm birth [12,14,15], reduced allergies and asthma in children [16,17], and improved neurocognitive outcomes in offspring [13]. In adults, higher consumption of n-3 fatty acids has been associated with cardio-protective effects such as anti-hyperlipidemia, anti-thrombotic, anti-inflammatory, anti-hypertensive, and anti-arrhythmic effects [8,18]. However, n-3 fatty acid intake in the Western diet is typically low compared to recommended intakes [14,15].

Furthermore, despite the health benefits associated with n-3 fatty acid intake during distinct life stages, gaps in knowledge remain with regard to how n-3 fatty acid intake differs by age and/or gender in representative samples of the US population. For instance, a recent study found that only 14.1% of children aged 7–12 years met the National Academy of Medicine (NAM; formerly known as the Institute of Medicine (IOM)) recommendations for DHA and EPA intake for their age and gender [19]. Similarly, a study performed in Australia demonstrated that adolescents (12–17 years of age) continued to fall below Australian National Heart Foundation recommended intakes (≥500 mg/day EPA+DHA) despite significant increases in intake that occurred with age [20]. Evidence regarding n-3 fatty acid intake in early childhood is also lacking, which has particularly important developmental implications. 

Previous research has also provided preliminary insights into n-3 fatty acid intake in other subgroups of the total population. With regard to pregnancy, our previous analysis of the National Health and Nutrition Examination Survey (NHANES) data from 2003–2012 demonstrated that pregnant women and women of childbearing age have lower intakes of n-3 fatty acids compared to males of the same age [21]. A subsequent NHANES analysis using data from 2001–2014 found that 100% of childbearing-age women and pregnant women did not meet the 8 ounce/week seafood consumption recommendation in the 2015–2020 Dietary Guidelines for Americans, with over 95% of women in this age range not meeting the EPA+DHA intake recommendation of 250 mg/day [22]. 

In order to identify subgroups of the US population at risk of low omega-3 fatty acid intake, the objective of this study was to assess current n-3 fatty acid intake in subgroups of the US population and evaluate differences based on age, gender, and pregnancy status using NHANES data as a nationally representative sample. Fish intake and supplement use were also analyzed in these groups.

## 2. Materials and Methods 

NHANES is a cross-sectional survey conducted by the National Center for Health Statistics (NCHS), under the Centers for Disease Control and Prevention, using a complex multistage probability sample that is designed to be representative of the national civilian US population [23]. Sampling weights were adjusted to account for multiple cycles. Males and females aged 1 year or older were included. Adult men with caloric intakes of <800 kcal or >8000 kcal per day and adult women with caloric intakes of <600 kcal or >6000 kcal per day were excluded. Daily intakes of the long-chain n-3 fatty acids eicosapentaenoic acid (EPA) and docosahexaenoic acid (DHA) were calculated for the total US population and the following age groups: toddler/early childhood (1–5 years), middle childhood (6–11 years), adolescents (12–19 years), adults (20–55 years) and seniors (55+ years). Each age group was also analyzed by gender. 

Population mean daily intakes of EPA and DHA are reported from two averaged 24-h dietary recall interviews from each participant. Trained dietary interviewers collected detailed information on all foods and beverages consumed by respondents in the previous 24-h time period (midnight to midnight). A second dietary recall was administered by telephone 3 to 10 days after the first dietary interview, but not on the same day of the week as the first interview. If an individual did not complete the second dietary interview, only the first dietary interview was used for the EPA and DHA intake from food. 

Participants were also asked if they had taken a dietary supplement in the past 30 days, how long they had been taking it, how many days it was taken in the past 30 days, the amount that was taken on those days, and the reason(s) that they were taking it. Label information, such as supplement name, manufacturer and/or distributor, serving size, form of serving size, and ingredients, were recorded for each supplement reported by participants. For each supplement, the amount of EPA and DHA provided was obtained from the supplement label. When the EPA and DHA content was not specified on the supplement label, the EPA and DHA content was imputed based on the proportion of EPA and DHA in the n-3 fatty acid-containing ingredient (i.e., 18% EPA and 12% DHA per 1 g of fish oil; 8% EPA and 10% DHA per 1 g of cod liver oil; and 8% EPA and 12% DHA per 1 g of salmon oil). Only supplements containing fish oil, cod liver oil, salmon oil, krill oil, and DHA-only preparations were included. 

Total fish intake in the last 30 days was calculated based on the number of times a respondent reported eating each of the 31 types of fish included in the NHANES survey over the last 30 days. If an individual answered “No” for eating a particular fish, the number of times that particular fish was consumed in the last 30 days was assumed to be zero. Additionally, intake of fish high in n-3 fatty acids was calculated using the above described method but limited to 7 fish (tuna, mackerel, salmon, sardines, shark, swordfish, and trout). 

Descriptive statistics (counts and percentages, means and standard errors and confidence intervals) are shown for all participants. SAS version 9.4 (SAS Institute Inc., Cary, NC, USA) was used for all statistical analyses. SURVEYFREQ, SURVEYMEANS, and SURVEYREG were used to compute descriptive statistics and regression analyses as these procedures account for complex survey design and sampling weight. The Rao–Scott chi-square test was used to assess the association between categorical variables. Continuous variables were compared using regression analysis (SURVEYREG). All comparisons included an adjustment for energy intake. Pairwise comparisons between groups were adjusted for multiple comparisons using the Bonferroni method. Significance was set at *p* < 0.05.

When assessing differences between IOM/World Health Organization (WHO) and NHANES age groups, the recommended intakes for the age groups of interest were averaged to use as a reference with the observed NHANES data. For example, in comparing our early childhood age group (1–5 year) with IOM recommendations, we utilized 150 mg/day as an average of the suggested intakes for the 2–4 years (100–150 mg/day) and 4–6 years (150–200 mg/day) WHO age classifications. Further, the IOM does not give direct recommendations for EPA+DHA; instead the IOM recommends 10% of the total dietary intake of n-3 fatty acids comes from EPA+DHA. 

## 3. Results

### 3.1. Participant Characteristics

Participant characteristics (*n* = 45,347) are provided in Table 1, including average EPA, DHA, and EPA+DHA intakes across survey participants/subgroups and the number of participants included in each subgroup.

### 3.2. EPA and DHA Intake by Age, Gender, and Pregnancy Status 

EPA and DHA intake by age group is shown in Table 2. Adults and seniors had significantly higher intakes of EPA, DHA, and EPA+DHA per 1000 kcal (53.5 mg EPA+DHA per 1000 kcal and 66.7 mg EPA+DHA per 1000 kcal, respectively) compared to adolescents (29.7 mg EPA+DHA per 1000 kcal; *p* < 0.001). EPA+DHA intake adjusted for caloric intake (mg EPA+DHA per 1000 kcal) was higher in adolescents compared to toddlers/early childhood (23.1 mg EPA+DHA per 1000 kcal; *p* < 0.001). Adults and seniors had equivalent intake (*p* = 1.0) of EPA and DHA, as well as their combination per 1000 kcal. Regarding gender, intake of EPA, DHA, and combined EPA+DHA was significantly higher in males than females, overall and within each age group, (*p* < 0.001 for all n-3 fatty acids/combinations) with intake by men of all age groups being consistently higher than females of the same age group (Table 3). There were no significant differences in EPA, DHA, or EPA+DHA intake in pregnant women compared to non-pregnant women of childbearing age (100.6 vs. 92.7 mg EPA+DHA in pregnant versus non-pregnant women aged 20–44 years; Table 4). Together, these data indicate significant differences in n-3 fatty acid consumption across age subgroups for the years surveyed, suggesting that younger individuals may be at risk for low n-3 fatty acid intake. Furthermore, women may be at particular risk for low n-3 fatty acid intake compared to men.

### 3.3. Fish Intake by Age Group, Gender, and Pregnancy Status

Intakes of total and high n-3 fatty acid-containing fish over a 30-day period are presented in Table 5, based on age group, gender, and pregnancy status. Similar to our findings for total EPA, DHA, and EPA+DHA intake, there was a significant difference in the intake of total fish and fish high in n-3 fatty acids based on age. Pairwise comparisons identified that adults and seniors consumed significantly more total fish (6.5 servings/month for both) and fish high in n-3 fatty acids (2.6 servings/month for both) compared to the toddler/early childhood (4 and 1.5 servings/month, respectively), middle childhood (3.8 and 1.5 servings/month, respectively), and adolescent age groups (3.9 and 1.6 servings/month, respectively; *p* < 0.001 for all comparisons). Intake of total and n-3 fatty acid-rich fish did not differ significantly amongst the adolescent, middle childhood, or toddler/early childhood groups, nor did it differ between the adults and seniors. These findings concur with the 24-h dietary recall findings indicating significantly lower consumption of n-3 fatty acid-containing foods (i.e., fish) by younger individuals.

Differences in fish consumption by age also remained when comparing male and female age subgroups (Table 6). Overall, males consumed approximately 5% more servings of total fish compared to females (6.1 vs. 5.8 servings of total fish/month, respectively; *p* < 0.001), whereas women consumed about 4% more servings per month of fish high in n-3 fatty acids than men (2.4 vs. 2.3 servings/month, respectively; *p* = 0.006). When comparing pregnant women versus non-pregnant women of childbearing age (Table 5), pregnant women reported consuming 1.8 servings/month of high n-3 fatty acid-containing fish, which was significantly lower than the 2.6 servings/month reported by non-pregnant women (*p* < 0.001). Findings were similarly lower for total fish intake in pregnant women compared to women of childbearing age. These results further corroborate the findings of our analysis of the 24-h dietary recall data, whereby younger individuals and females report lower consumption of n-3 fatty acid-containing foods (i.e., fish) compared to older individuals and men.

### 3.4. EPA/DHA-Containing Supplement Use by Age, Gender, and Pregnancy Status

The total percentage of participants taking EPA/DHA-containing supplements was 0.8% (Table 7). EPA, DHA, and combined EPA+DHA intake was significantly higher in individuals reporting use of any supplement (with or without n-3 fatty acids) compared to individuals who did not report supplement use (Table 8; *p* < 0.001). Individuals taking an EPA/DHA-containing supplement had significantly elevated intake compared to individuals not taking n-3 fatty acid-containing supplements or not reporting any supplement use (Table 8; *p* < 0.001). While 0.6% of childbearing-age women reported taking an EPA/DHA-containing supplement, 7.3% of pregnant women reported use of EPA/DHA-containing supplements. As supplement use is associated with increased n-3 fatty acid intake, supplementation could be an important source of EPA/DHA, particularly for pregnant women given their lower fish consumption compared to non-pregnant women of childbearing age.

## 4. Discussion

Our analysis of the NHANES 2003–2014 data demonstrates that there are significant differences in EPA and DHA intake based on age, gender, and pregnancy status, with potentially vulnerable populations (i.e., children and women) consuming amounts that are well below recommended intake levels. Overall, intake of EPA, DHA, EPA+DHA, and fish intake (both total and n-3 fatty acid-rich fish) increased with age. Only 0.8% of the study population reported using EPA/DHA supplements, but this was associated with a significantly higher EPA, DHA, and EPA+DHA intake compared to non-supplement users; thus, EPA/DHA supplements may provide a potential alternative or complement to dietary sources (e.g., cold-water fatty fish such as salmon, mackerel, tuna, herring, and sardines) and an effective means of increasing n-3 fatty acid intake. Current recommendations for daily n-3 fatty acid intake vary widely and recommended daily allowances or dietary reference intakes have not been set for DHA and EPA across age groups. However, numerous groups have issued recommendations for adequate intake (AI) of n-3 fatty acids based on age, health, and other factors (Table 9). In our analysis, all groups across age, gender, and pregnancy status failed to meet these recommended intakes for EPA, DHA, and EPA+DHA, putting vulnerable populations at potential risk for adverse health outcomes. 

Although the lack of standardized recommendations makes direct comparisons challenging, results consistently indicate that individuals from 1–19 years of age have low n-3 fatty acid intakes. Low n-3 fatty acid intake during this stage of life has significant implications given the role of these fatty acids in the development of the central nervous system and retina [12]. Pre- and postnatal intake of n-3 fatty acids in infants has been associated with improved neurocognitive outcomes [13], higher birth weights and lengths [30,31,32]. In early childhood, n-3 fatty acids are a part of maintenance and cell turnover throughout the body and have been shown to have a positive impact on brain activity, learning and cognition [27,28,29,33]. In children 1–5 years old, we found that total EPA+DHA intake (32.4 mg/day or 23.1 mg per 1000 kcal) reached only ~40% of the EPA+DHA intake recommended by the IOM for age groups 1–3 years (70 mg/day) and 4–8 years (90 mg/day). When compared to the WHO recommendations, the toddler/early childhood age group reached less than one-quarter of the recommended intake of ~150 mg/day. This corroborates findings from a 2013 study in which toddlers aged 13–24 months had inadequate intake of n-3 fatty acids according to IOM dietary recommendations [34]. Children aged 6–11 years old also demonstrated low intakes of n-3 fatty acids according to the IOM and WHO recommendations with EPA+DHA intakes (47.6 mg/day or 25.9 mg per 1000 kcal) ranging from ~4.5–27% of the IOM’s (950–1050 mg) and WHO’s (175–225 mg) recommended daily intakes, respectively. Similar results were found in the Southeastern United States, where only 14.1% of children aged 7–12 years reached IOM recommendations for EPA and DHA [19]. Adolescents (12–19 years) likewise had low n-3 fatty acid intake (59.2 mg/day EPA+DHA or 29.7 mg per 1000 kcal), reaching approximately 50% of the IOM recommendation for ages 14–18 years (160 mg/day for males and 110 mg/day for females) and falling well short of the WHO recommendation for all ages past 10 years (200–500 mg/day). Similarly, Gopinath et al. found that mean n-3 fatty acid intake remained low in adolescents aged 12 to 16 years old when compared to younger ages, despite a significant increase in energy-adjusted dietary intake of total n-3 fatty acids, EPA, and DHA [20]. 

The importance of n-3 fatty acids extends long past childhood as they also have vital physiological roles in adulthood and advanced age. For instance, eicosanoids and other bioactive lipid mediators derived from n-3 fatty acid metabolism are signaling molecules that have a variety of functions in the cardiovascular, pulmonary, immune and endocrine systems [35,36]. With regard to cardiovascular disease risk factors, there is strong evidence that n-3 fatty acids can beneficially modify blood lipids, inflammation, and endothelial function [2]. In older adults (55+ years old), n-3 fatty acids have also been shown to help modulate depression and cognitive decline [37]. Although adults and seniors had the highest EPA+DHA intakes in our study, they still fell well below recommended levels (Table 9). For instance, EPA+DHA intake in adults and seniors (53.3 and 66.7 mg/day per 1000 kcal, respectively) reached only 16% and 21% of the 325 mg/day per 1000 kcal minimum recommendation from the Workshop on the Essentiality of and Recommended Dietary Intake. Using the minimum recommendation of 200 mg/day of EPA+DHA given by many organizations, mean intake by adults and seniors in our study (with unadjusted daily intakes of 112.5 mg and 118.5 mg EPA+DHA, respectively) met less than 60% of the intake recommendation. 

Despite the distinct functional roles n-3 fatty acids play across the lifespan, most recommendations do not address older adults specifically. Although adolescent and elderly brains experience different changes, the IOM classifies these two populations together and recommends the same n-3 fatty acid intakes for ages 14–55+ years. Similarly, most organizations do not make a distinction between adults and seniors, despite known age-related changes in the elderly, and provide the same recommendations for 19–51+ years. Additional research evaluating the potential differing n-3 fatty acid physiological requirements across these age groups is needed for the development of evidence-based recommendations and may have implications for modulating risk factors to protect against chronic disease.

Gender and age disparities may also exist for n-3 fatty acid intake. In our analysis, men reported significantly higher n-3 fatty acid intake compared to women. In both men and women, adults and seniors had higher intakes compared to adolescents and children. Given the reduced intake in n-3 fatty acids in females compared to males, it is important to consider the potential implications for pregnancy due to fetal reliance on maternal DHA for brain and retinal development [4]. The sole source of n-3 fatty acids in breast-fed infants is the mother’s milk [38] and low n-3 polyunsaturated fatty acid intake during pregnancy and breast-feeding could be a contributing factor to poor infant neural development [30,32,33]. In the current study, we found significantly lower fish consumption in pregnant women compared to non-pregnant women of childbearing age, including reduced consumption of high n-3 fatty acid-containing fish. This may be due in part to concerns about the effects of mercury present in fish on their pregnancy and fetal development, causing women to decrease their consumption of fish or avoid it all together. Despite consistent findings of lower fish intake in pregnant women [21,22], we did not find a significant difference in total EPA, DHA, and EPA+DHA intake between pregnant women and non-pregnant women between 20–44 years of age. This may be explained in part by increased EPA/DHA-containing supplement use by pregnant women (7.3% compared to 0.6% in non-pregnant women of childbearing-age), which is consistent with our previous findings regarding supplement use [21]. Compared to the recommendations given by the Workshop of Essentiality of and Recommended Dietary Intake, estimated EPA+DHA intake by pregnant women in our study was less than one-fifth of the recommended 520 mg/day EPA+DHA for pregnant and lactating women [25], consistent with our previous studies [21,22]. Together, these studies indicate that low EPA+DHA is a persistent problem in pregnant women in the US. 

### Strengths and Limitations

We analyzed a large, representative sample of the US population, stratifying by age, gender, and pregnancy status. Our EPA+DHA intake findings are based on a 24-h dietary recall method that is considered sufficient for accurately measuring mean dietary intakes at the population level [39]. However, this may not reflect an individual’s habitual dietary pattern, and a recent study identified discrepancies in the NHANES dataset with regard to energy intake values within plausible ranges [40]. To reduce the potential effect of this, we limited our study only to participants who reported plausible caloric intake ranges and excluded adult men with caloric intakes of <800 kcal or >8000 kcal per day and adult women with caloric intakes of <600 kcal or >6000 kcal per day. Furthermore, in addition to the 24-h dietary recall analysis, we also included an assessment of fish intake over a 30-day period that corroborated and expanded upon our overall EPA+DHA intake findings. However, it should also be noted that assessing n-3 fatty acid intake based on fish intake also has limitations. For instance, the EPA and DHA of a particular species of fish can differ based on its source, with some species of farmed fish having substantially lower EPA and DHA than their wild-caught counterparts [41]. Nonetheless, our results are in agreement with other recent investigations, providing consistent evidence for interpretation [19,20,21,42]. The cross-sectional nature of NHANES must also be considered in the interpretation of our results as we cannot establish any causal relationships related to intake. The lack of standardized guidelines for sufficient n-3 fatty acid intake also impedes our ability to draw inferences from the current results. In particular, guidelines that are specific for age, gender, or vulnerable subgroup populations are limited, and inadequate research into the required levels of n-3 fatty acids to support specific physiological functions presents a hindrance to developing strong, evidence-based recommendations. Furthermore, our study does not assess for other factors that may influence results (e.g., geographic regions, ethnicities, or socioeconomic factors) and these warrant further consideration in future studies.

## 5. Conclusions

To our knowledge, this is the first analysis of the 2003–2014 NHANES data to assess for differences in EPA+DHA intake across the lifespan and between genders. We found that n-3 fatty acid intake across all age groups was lower than recommended amounts. Within age subgroups, younger individuals have lower DHA+EPA intake and fish consumption compared to adults and seniors. Furthermore, women had lower n-3 fatty acid intake compared to men across age groups, with no difference in EPA+DHA intake between pregnant women and non-pregnant women of childbearing age. Taken together, these findings demonstrate that low n-3 fatty acid intake is consistent among the US population and could increase the risk for adverse health outcomes, particularly in vulnerable populations (e.g., young children and pregnant women).

## Figures and Tables

**Table 1 nutrients-11-00177-t001:** Descriptive characteristics of the National Health and Nutrition Examination Survey (NHANES) study population (*n* = 45,347).

Characteristics		*n*	Mean (SE ^1^)
	Age (years)	45,347	37.2 (0.3)
	EPA ^2^ intake (mg)	45,244	32.6 (1.0)
	DHA ^3^ intake (mg)	45,244	64.4 (1.5)
	EPA+DHA intake (mg)	45,244	96.9 (2.5)
	Total fish intake (servings/month)	33,403	6.5 (0.1)
	Fish high in n-3 intake (servings/month)	18,562	2.5 (0.1)
			**(%)**
Gender	Male	22,056	48.5
	Female	23,291	51.5
Pregnant	Yes	762	4.8
	No	8381	92.1
	Do not know	287	3.1
Age group	Toddler/Early Childhood (1–5 years)	5495	6.9
	Middle Childhood (6–11 years)	5550	8.3
	Adolescent (12–19 years)	8186	11.5
	Adult (20–55 years)	15,937	50.2
	Seniors (55+ years)	10,179	23.1

^1^ SE, standard error; ^2^ EPA, eicosapentaenoic acid; ^3^ DHA, docosahexaenoic acid.

**Table 2 nutrients-11-00177-t002:** n-3 fatty acid (mg) per 1000 kcal intake by age group ^1^.

Age Group	Variable	*n*	Mean (SE ^2^), mg	95% CI for Mean
Toddler/Early Childhood 1–5 years	EPA	5392	6.9 (0.4)	6.2, 7.7
	DHA	5392	16.1 (0.6)	14.9, 17.4
	EPA+DHA	5392	23.1 (1.0)	21.1, 25.0
Middle Childhood 6–11 years	EPA	5550	8.4 (0.5)	7.4, 9.4
	DHA	5550	17.5 (0.9)	15.7, 19.2
	EPA+DHA	5550	25.9 (1.4)	23.1, 28.6
Adolescents 12–19 years	EPA	8186	9.9 (0.6)	8.8, 11.0
	DHA	8186	19.8 (0.9)	18.0, 21.5
	EPA+DHA	8186	29.7 (1.4)	26.9, 32.5
Adults 20–55 years	EPA	15,937	17.9 (0.6)	16.7, 19.1
	DHA	15,937	35.4 (0.9)	33.5, 37.2
	EPA+DHA	15,937	53.3 (1.5)	50.3, 56.3
Seniors 55 years & above	EPA	10,179	22.4 (0.9)	20.6, 24.2
	DHA	10,179	44.3 (1.6)	41.0, 47.5
	EPA+DHA	10,179	66.7 (2.5)	61.7, 71.7

^1^*p*-value < 0.001 for EPA, DHA, and EPA+DHA intake between age groups adjusted for energy intake. ^2^ SE, standard error.

**Table 3 nutrients-11-00177-t003:** n-3 fatty acid intake (mg) per 1000 kcal by gender and age subgroup.

Gender	Variable	Adolescents 12–19 years	Adults 20–55 years	Seniors 55+ years	*p*-Value
		(*n* = 8203)	(*n* = 16,215)	(*n* = 10,475)	
		Mean (SE ^1^), mg	Mean (SE ^1^), mg	Mean (SE ^1^), mg	
Male	EPA	9.3 (0.6)	17.6 (0.7)	22.2 (1.3)	<0.001
	DHA	19.1 (1.0)	34.6 (1.0)	43.7 (2.5)	<0.001
	EPA+DHA	28.4 (1.6)	52.1 (1.7)	65.9 (3.8)	<0.001
Female	EPA	10.5 (0.8)	18.3 (0.9)	22.5 (1.3)	<0.001
	DHA	20.4 (1.1)	36.1 (1.5)	44.8 (2.0)	<0.001
	EPA+DHA	31.0 (1.8)	54.4 (2.4)	67.3 (3.2)	<0.001

^1^ SE, standard error.

**Table 4 nutrients-11-00177-t004:** n-3 fatty acid intake by pregnancy status.

	Pregnant Women ^1^	Non-Pregnant Women	*p*-Value
	(*n* = 762, 8.3%)	(*n* = 8381, 91.7%)	
	Mean (SE ^2^), mg	Mean (SE ^2^), mg	
EPA	33.1 (4.1)	31.6 (1.9)	0.76
DHA	67.5 (11.8)	61.1 (2.8)	0.60
EPA+DHA	100.6 (15.2)	92.7 (4.7)	0.63

^1^ Pregnancy status applied to women of age 20–44; 287 women had no status; ^2^ SE, standard error.

**Table 5 nutrients-11-00177-t005:** Total fish intake (servings/month) and intake of fish high in n-3 fatty acids for a 30-day period.

	Total Fish				High n-3 Fish			
	*n*	Mean (SE ^1^)	95% CI	*p*-Value	*n*	Mean (SE ^1^)	95% CI	*p*-Value
**Age Group**								
Toddler/Early Childhood	4040	4 (0.1)	(3.7, 4.2)		3427	1.5 (0.1)	(1.3, 1.6)	
Middle Childhood	3514	3.8 (0.1)	(3.6, 4.1)		2859	1.5 (0.1)	(1.3, 1.7)	
Adolescent	4608	3.9 (0.1)	(3.6, 4.2)		3344	1.6 (0.1)	(1.4, 1.8)	
Adult	13,395	6.5 (0.1)	(6.2, 6.7)		11,584	2.6 (0.1)	(2.5, 2.7)	
Senior	7846	6.5 (0.1)	(6.2, 6.8)	<0.001	7348	2.6 (0.1)	(2.4, 2.7)	<0.001
**Gender**								
Male	15,829	6.1 (0.1)	(5.8, 6.4)		13,519	2.3 (0.1)	(2.2, 2.5)	
Female	17,574	5.8 (0.1)	(5.6, 6)	<0.001	15,043	2.4 (0.1)	(2.3, 2.6)	0.006
**Pregnant**								
Yes	630	4.6 (0.2)	(4.1, 5.1)		511	1.8 (0.2)	(1.4, 2.1)	
No	6276	6.1 (0.1)	(5.8, 6.4)		5279	2.6 (0.1)	(2.4, 2.8)	
Cannot Determine	171	5.9 (0.7)	(4.6, 7.2)	<0.001	141	2.6 (0.3)	(2.1, 3.2)	<0.001

^1^ SE, standard error.

**Table 6 nutrients-11-00177-t006:** Fish intake by age in males and females over a 30-day period.

	Male				Female			
	*n*	Mean (SE ^1^)	95% CI	*p*-Value	*n*	Mean (SE ^1^)	95% CI	*p*-Value
**Total Fish**								
Adolescent	2182	3.9 (0.2)	(3.6, 4.3)		2426	3.9 (0.2)	(3.5, 4.2)	
Adult	5994	6.8 (0.2)	(6.4, 7.1)		7401	6.2 (0.1)	(5.9, 6.5)	
Senior	3865	6.7 (0.2)	(6.4, 7.1)	<0.001	3981	6.3 (0.2)	(6, 6.6)	<0.001
**High n-3 Fish**								
Adolescent	1599	1.5 (0.1)	(1.3, 1.7)		1745	1.6 (0.1)	(1.4, 1.9)	
Adult	5160	2.7 (0.1)	(2.5, 2.8)		6424	2.6 (0.1)	(2.4, 2.7)	
Senior	3597	2.4 (0.1)	(2.2, 2.6)	<0.001	3751	2.8 (0.1)	(2.6, 3)	<0.001

^1^ SE, standard error.

**Table 7 nutrients-11-00177-t007:** Proportion of participants taking a supplement containing EPA/DHA.

	Reported EPA/DHA Supplement Use (%)
**Total Population**	328 (0.8)
**Age**	
Toddler/Early Childhood	26 (0.5)
Middle Childhood	36 (0.6)
Adolescent	26 (0.4)
Adult	139 (0.9)
Senior	101 (1.0)
**Gender**	
Male	177 (0.8)
Female	151 (0.8)
**Pregnant**	
Yes	23 (7.3)
No	41 (0.6)

**Table 8 nutrients-11-00177-t008:** EPA/DHA intake categorized by supplement use.

	EPA			DHA		
	Mean (SE ^1^), mg	95% CI	*p*-Value	Mean (SE ^1^), mg	95% CI	*p*-Value
**Total Population**						
Without supplement information	32.6 (1.0)	30.7, 34.5		64.4 (1.6)	61.3, 67.4	
With supplement information	33.9 (1.0)	32.0, 35.9	<0.001	65.7 (1.6)	62.6, 68.8	<0.001
**By Supplement Type**						
Supplement with n-3	207.6 (16.1)	175.6, 239.7		240.1 (11.9)	216.5, 263.8	
Supplement w/o n-3	35.1 (1.3)	32.5, 37.8		68.8 (2.2)	64.5, 73.1	
Does not take supplement	29.7 (1.0)	27.6, 31.7	<0.001	59.3 (1.5)	56.2, 62.3	<0.001

^1^ SE, standard error.

**Table 9 nutrients-11-00177-t009:** n-3 fatty acid dietary reference intakes.

World Health Organization [24]		
	**Age**	**Adequate Intake (AI):**
DHA	12–24 months	10–12 mg/kg
EPA+DHA	2–4 years	100–150 mg
	4–6 years	150–200 mg
	6–10 years	200–250 mg
	Adults	200–500 mg
National Institute of Medicine [11]		
	**Age**	**Male/Female as (mg/day)**
EPA+DHA *	1–3 years	70 mg/70 mg
	4–8 years	90 mg/90 mg
	9–13 years	120 mg/100 mg
	14–18 years	160 mg/110 mg
	19–50 years	160 mg/110 mg
	51+ years	160 mg/110 mg
Workshop on the Essentiality of and Recommended Dietary Intakes for n-6 and n-3 fatty acids ** [25]		
	**Adult intake based on a 2000 kcal diet**	
DHA	220 mg/day	
EPA	220 mg/day	
EPA+DHA	650 mg/day	
Pregnancy and Lactation		
DHA	300 mg/day	
EPA	220 mg/day	
American Heart Association		
Adults [26]	Eat at least 8 oz of fish/week (equal to 2 servings/week	
U.S. Dept of Agriculture and U.S. Department of Health and Human Services [27]	Adults	Increase the amount and variety of seafood consumed, in place of some meat and poultry
	Pregnant or Breastfeeding Women	Consume at least 8 and up to 12 ounces per week of a variety of seafood
FDA [28]	For women of childbearing age (approximately 16–49 years old), especially pregnant and breastfeeding women, and for parents and caregivers of young children. Eat 2 to 3 servings of fish a week from the “Best Choice” list (i.e., herring, shrimp, tilapia) OR 1 serving from the “Good Choice” list (i.e., snapper, halibut, Mahi mahi). Eat a variety of fish. Serve 1 to 2 servings of fish a week to children, starting at age 2.	
The American College of Obstetricians and Gynecologist [29]	Eat at least 2 servings of fish or shellfish (8–12 oz) per week before becoming pregnant, while pregnant, and while breastfeeding.	

* The National Institute of Medicine suggests 10% of ALA intake be from EPA+DHA, with no specific intake recommendations for EPA or DHA. Values in this table were calculated from recommended ALA values. ** Workshop sponsored by National Institute on Alcohol Abuse and Alcoholism-NIH, the Office of Dietary Supplements-NIH, The Center for Genetics, Nutrition and Health, and the International Society for the Study of Fatty Acids and Lipids.

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
