# Peer review of "Omega-3 Fatty Acid Intake by Age, Gender, and Pregnancy Status in the United States: National Health and Nutrition Examination Survey 2003–2014"

_nutrients, 2019, doi:10.3390/nu11010177_

Reviewer 1 Report

Excellent manuscript which will provide a valuable resource for individuals working in this area.  Except for a few grammatical errors.

Author Response

We thank the reviewer for their positive feedback on our manuscript, and have edited the manuscript for improved grammatical clarity.

Reviewer 2 Report

The following suggestions may increase the merit of the manuscript. 1. The General Introduction section needs to be structured better to provide the specific Background pertaining to the study and to establish the platform for the rationale of the study. 2. The Results are enunciated in a very descriptive manner. The Results need to be described in a relatively more detailed manner with greater depth and added insight. 3. In the Discussion section, the authors must highlight the novelty and significance of the findings and observations emanating from the study.

Author Response

The following suggestions may increase the merit of the manuscript.

1. The General Introduction section needs to be structured better to provide the specific Background pertaining to the study and to establish the platform for the rationale of the study.

RESPONSE: Thank you for this suggestion. We have modified the introduction as requested to better frame the rationale for this study.

2. The Results are enunciated in a very descriptive manner. The Results need to be described in a relatively more detailed manner with greater depth and added insight.

RESPONSE: We appreciate this suggestion. To address this concern, we have added greater detail/depth to the results section of the manuscript as suggested.

3. In the Discussion section, the authors must highlight the novelty and significance of the findings and observations emanating from the study.

RESPONSE: Thank you for raising this concern. We have modified the manuscript to include a conclusions section to better highlight the significance of the findings described in the manuscript.

Reviewer 3 Report

Thompson et al report data from the NHANES study, generated in the years 2003 – 2014, on omega-3 fatty acid intake, and group the data reported by age and gender. NHANES is a well-known national cross-sectional survey, designed to be representative for the US population. Age groups formed were “toddler/early childhood (1 – 5 y), middle childhood (6 – 11 y), adolescents (12 – 19 y), adults (20- 55 y) and seniors (55+ y).” Omega-3 intake was assessed by two 24 hour recall interviews from each participant, one in person, and one by phone. Participants were also questioned on intake of supplements containing EPA and DHA in the last 30 days, which was quantified by use of typical concentrations in the various supplements, and the amounts reported to have been taken. Similarly, intake of omega-3 fatty acids via fish was quantified. It was found that younger individuals reported lower ingestion of omega-3 fatty acids than older individuals, and females reporting lower intakes than males, also after adjustment for caloric intake. Women reported to have consumed less fish than men, and pregnant women reported no more intake of fish than non-pregnant women. The authors conclude “Our findings indicate that subgroups of the population may be at higher risk for suboptimal n-3 fatty acid intakes.”

Introduction is of adequate length, and develops the question properly. Methods and Results are clearly described. Discussion is lengthy, and does not mention the major shortcomings of the study.

Major Points

A plausibility check on the data generated by memory-based methods of assessing dietary intake in NHANES found that only 50% of data thus generated are plausible (1). This is especially true for food items not ingested daily. Therefore, the quality of the data thus generated is questionable.

Omega-3 intake from fish was calculated, but Omega-3 content of farmed fish, which comprises a substantial portion of fish ingested is far from constant – e.g. farmed salmon contained some 50% less omega-3 fatty acids in 2006 than in 2016 (2). Therefore, also these calculations come with a margin of error of some 50%. Therefore, the quality of the data thus generated is also questionable.

Moreover, the conclusion drawn is grossly inadequate. An optimal intake of omega-3 fatty acids has never been defined, rather recommendations on amounts to be eaten have been given. These recommendations vary from 200 mg to 1 g EPA + DHA per day, i.e. by a factor of 5 – reflecting at least substantial disagreement. An optimal intake has never been defined, and therefore it is unknown what a “suboptimal intake” is.

1. Archer E, Hand GA, Blair SN. Validity of U.S. nutritional surveillance: National Health and Nutrition Examination Survey caloric energy intake data, 1971-2010. PLoS One. 2013 Oct 9;8(10):e76632.

2. Sprague M, Dick JR, Tocher DR. Impact of sustainable feeds on omega-3 long-chain fatty acid levels in farmed Atlantic salmon, 2006-2015. Sci Rep. 2016 Feb 22;6:21892.

Author Response

1) A plausibility check on the data generated by memory-based methods of assessing dietary intake in NHANES found that only 50% of data thus generated are plausible (reference 1). This is especially true for food items not ingested daily. Therefore, the quality of the data thus generated is questionable.

RESPONSE: Thank you for raising this concern. We have modified the manuscript to include a discussion of this limitation as a consideration in evaluating the results of this study. Additionally, we limited our study to survey participants that reported plausible caloric intake ranges (adult men with caloric intakes of < 800 kcal or > 8000 kcal per day and adult women with caloric intakes of < 600 kcal or > 6000 kcal per day were excluded) in order to mitigate this potential limitation.  We also analyzed both calculated n-3 fatty acid intake (based on 24-hr recall data) in addition to separately assessing 30-day fish intake, in an attempt to compensate for inaccuracies that could occur due to survey bias for food items not ingested daily.

2) Omega-3 intake from fish was calculated, but Omega-3 content of farmed fish, which comprises a substantial portion of fish ingested is far from constant – e.g. farmed salmon contained some 50% less omega-3 fatty acids in 2006 than in 2016 (reference 2). Therefore, also these calculations come with a margin of error of some 50%. Therefore, the quality of the data thus generated is also questionable.

RESPONSE: We recognize the validity of the reviewer’s concern over discrepancies in wild-caught versus farmed fish in assessing omega-3 fatty acid content of a particular fish species. We have modified the manuscript to include a discussion of this limitation and the potential effect of this variability in determining omega-3 fatty acid content from fish intake without specifying the source of fish.

3) Moreover, the conclusion drawn is grossly inadequate. An optimal intake of omega-3 fatty acids has never been defined, rather recommendations on amounts to be eaten have been given. These recommendations vary from 200 mg to 1 g EPA + DHA per day, i.e. by a factor of 5 – reflecting at least substantial disagreement. An optimal intake has never been defined, and therefore it is unknown what a “suboptimal intake” is.

RESPONSE: Thank you for raising this concern. We have revised the manuscript to refrain from using the term “suboptimal intake” or terms relating to sufficiency/insufficiency.

Round  2

Reviewer 3 Report

no further comments.